# Identification of a Common Epitope in Nucleocapsid Proteins of Euro-America Orthotospoviruses and Its Application for Tagging Proteins

**DOI:** 10.3390/ijms22168583

**Published:** 2021-08-10

**Authors:** Hao-Wen Cheng, Wei-Ting Tsai, Yi-Ying Hsieh, Kuan-Chun Chen, Shyi-Dong Yeh

**Affiliations:** 1Department of Plant Pathology, National Chung-Hsing University, Taichung 40227, Taiwan; cashowen@gmail.com (H.-W.C.); Livelytracy1014@gmail.com (W.-T.T.); sky62800@hotmail.com (Y.-Y.H.); bact.tw@yahoo.com.tw (K.-C.C.); 2Advanced Plant Biotechnology Center, National Chung-Hsing University, Taichung 40227, Taiwan

**Keywords:** orthotospovirus, epitope tag, nucleocapsid protein

## Abstract

The NSs protein and the nucleocapsid protein (NP) of orthotospoviruses are the major targets for serological detection and diagnosis. A common epitope of KFTMHNQIF in the NSs proteins of Asia orthotospoviruses has been applied as an epitope tag (nss-tag) for monitoring recombinant proteins. In this study, a monoclonal antibody TNP MAb against the tomato spotted wilt virus (TSWV) NP that reacts with TSWV-serogroup members of Euro-America orthotospoviruses was produced. By truncation and deletion analyses of TSWV NP, the common epitope of KGKEYA was identified and designated as the np sequence. The np sequence was successfully utilized as an epitope tag (np-tag) to monitor various proteins, including the green fluorescence protein, the coat protein of the zucchini yellow mosaic virus, and the dust mite chimeric allergen Dp25, in a bacterial expression system. The np-tag was also applied to investigate the protein–protein interaction in immunoprecipitation. In addition, when the np-tag and the nss-tag were simultaneously attached at different termini of the expressed recombinant proteins, they reacted with the corresponding MAbs with high sensitivity. Here, we demonstrated that the np sequence and TNP MAb can be effectively applied for tagging and detecting proteins and can be coupled with the nss-tag to form a novel epitope-tagging system for investigating protein–protein interactions.

## 1. Introduction

Viruses classified in the genus Orthotospovirus of the family Tospoviridae have enveloped quasi-spherical particles of 80–120 nm diameter, which contain three genomic single-stranded RNAs of negative sense large (L) RNA and ambisense medium (M) and small (S) RNAs [1,2,3]. The virus complementary (vc) sense of L RNA encodes the replicase L protein. The viral (v) sense M RNA encodes a nonstructural movement NSm protein and vc M RNA encodes Gn and Gc envelope proteins. S RNA encodes the nonstructural NSs protein from the v sense, and the structural nucleocapsid protein (NP) from the vc sense. The NSs protein is a gene silencing suppressor that antagonizes the defense reaction of RNA silencing in host plants [4,5,6]. The NP is a ssRNA binding protein and functions through the encapsidation of genomic RNA segments inside the envelope membrane, maintaining and stabilizing each genomic RNA molecule with the panhandle structure formed by the 5′ and 3′ complementary sequences [7]. The NP of the tomato spotted wilt virus (TSWV) has a homotypic interaction to form oligomers [8,9,10,11] and also interacts with Gn and Gc proteins in vivo [12,13].

Orthotospoviruses infect broad plant hosts and are distributed worldwide, especially in tropical and subtropical areas. They are transmitted by 14 species of thrips in nature [2]. The orthotospoviral NSs protein and NP are abundant and stable in the infected tissues; thus, they are the major targets of antiserum preparation for detecting and diagnosing orthotospoviruses. The demarcation of the genus *Orthotospovirus* into species is mainly based on the sequence comparison and serological relationship of their structural NPs. A threshold of 90% amino acid (aa) related to the identity of NPs has been proposed to classify orthotospoviruses at the species level [14,15]. So far, there are 31 species of orthotospoviruses that have been reported [16]. Based on the serological and phylogenetic analyses of NPs, all orthotospovirus species have been identified and classified into five serogroups, with TSWV, watermelon silver mottle virus (WSMoV), iris yellow spot virus (IYSV), groundnut yellow spot virus (GYSV), and soybean vein necrosis virus (SVNV) as type members; and two distinct lisianthus necrotic ringspot virus (LNRV) and impatiens necrotic spot virus (INSV) serotypes [15,17,18]. According to their geographic distribution, the TSWV and SVNV serogroups and the INSV serotype are classified as Euro-America type viruses, and the WSMoV, GYSV, and IYSV serogroups and the LNRV serotype are classified as Asia type viruses [19].

Many proteins with different therapeutic, diagnostic, or industrial purposes have been produced through various recombinant techniques with the modern advances in genomics, proteomics, and bioinformatics. Different expression hosts such as bacteria [20], yeast [21], plant [22], insect [23], and mammalian cell lines [24] have been developed to express recombinant proteins. Moreover, multiple tags are frequently used for stacking benefits including the detection, purification, solubilization, and immobilization of recombinant proteins [25,26,27,28]. Commonly used peptide tags include the polyhistidine-tag (his-tag) [29], the polyarginine-tag [30], the FLAG-tag [31], the Strep-tag [32], the c-Myc-tag [33], the HA-tag [34], the maltose-binding protein (MBP) [35], and glutathione-S-transferase (GST) [36].

An epitope is a small peptide sequence that a specific antibody can recognize on the same or different protein molecules. Using a small epitope tag is a better approach to minimize structural interference. The broad-spectrum diagnosis tool at the genus level of *Orthotospovirus* has been developed using several monoclonal antibodies (MAbs) against the common epitopes of NSs proteins to recognize Euro-America type or Asia type orthotospoviruses [19,37]. The nss-tag, a nine-aa oligopeptide of KFTMHNQIF, was derived from a highly conserved epitope present in NSs proteins of all WSMoV-serogroup and IYSV-serogroup orthotospoviruses [37]. The nss-tag has been successfully used for protein detection in the bacterial pET system and is applicable in transient expression by agroinfiltration in plants followed with co-immunoprecipitation to verify the protein–protein interactions in vitro [38]. Thus, the nss-tag is highly efficient for tagging recombinant proteins in bacterial and plant expression systems.

In this study, a monoclonal antibody MAb 20C4C8 against TSWV NP, designated as TNP MAb, that recognizes Euro-America orthotospoviruses, was produced. The MAb is an excellent tool for detecting TSWV-serogroup members and can distinguish them from other serogroup or serotype members of orthotospoviruses, especially Asia type viruses. The minimal length of the common epitope on TSWV NP recognized by TNP MAb was determined as having only five aa residues, ^211^KGKEY^215^. The six aa residues, ^211^KGKEYA^216^, designated as the np sequence, which has the full strength and high sensitivity of the reaction with TNP MAb, were used to test the feasibility of their role as an epitope tag in a bacterial expression system. The green fluorescence protein (GFP), ZYMV CP [38] and the chimeric house dust mite allergen (Dp25) [38] were fused with the np sequence at either the N- or C-terminus. All tagged proteins were expressed effectively in bacteria, and the reaction strength with TNP MAb was similar whether the np sequence was attached at the N- or C-extreme. The np sequence was also successfully applied to test the ZYMV CP and HC-Pro interaction by co-immunoprecipitation using TNP MAb. The nss-tag was also attached on all expressed recombinant proteins, and their detection and specificity did not interfere with the np sequence tag.

In conclusion, TNP MAb produced in this study is a valuable serological tool for detecting and diagnosing TSWV serogroup members of Euro-America orthotospoviruses. The np sequence, ^211^KGKEYA^216^ of TSWV NP, can be effectively applied in the bacteria protein expression system as an epitope tag, designated as the np-tag, and reacted with TNP MAb with high sensitivity. In addition, we demonstrate that the np-tag can be coupled with the nss-tag to form a novel epitope-tag system for monitoring protein expression and to study protein–protein interactions.

## 2. Results

### 2.1. TNP MAb 20C4C8 Recognizes the TSWV Serogroup of Euro-America Type Orthotospoviruses, but Not Asia Type Orthotospoviruses

The hybridoma cell line 20C4C8 was selected, and the produced ascitic fluid reacted with the TSWV-infected leaf tissue of *Nicotiana benthamiana* without background. An indirect ELISA was used to estimate the titer of TNP-MAb 20C4C8, which was designated as TNP MAb, and the titration endpoint was at 1,024,000× dilution. The results suggested that the best dilution for detection was 16,000× dilution, as the background reaction was low and the sensitivity was high at this dilution (Figure 1A).

Crude antigens from the *N. benthamiana* leaf tissue were individually infected with eight orthotospoviruses (TSWV, tomato chlorotic spot virus; TCSV, ground ringspot virus; GRSV, alstroemeria necrotic streak virus; ANSV, chrysanthemum stem necrotic virus; CSNV, WSMoV, INSV, or IYSV) were used to characterize the TNP MAb by western blotting. The TNP MAb reacted with the crude antigens of TSWV, TCSV, GRSV, ANSV, and CSNV, but not with those of INSV, WSMoV, and IYSV (Figure 1B). Our results indicated that the TNP MAb recognized the five members of the TSWV serogroup of Euro-America type orthotospoviruses, but not the other three orthotospoviruses of INSV, WSMoV, and IYSV. Thus, TNP MAb is an excellent tool for detecting TSWV-serogroup members and can distinguish them from other serogroup or serotype viruses, especially Asia type orthotospoviruses.

### 2.2. Identification of the Common Epitope Recognized by the TNP MAb 20C4C8

A conserved epitope was believed to be present in the NPs of Euro-America type orthotospoviruses since the tested members of this type were all well recognized by TNP (Figure 1). Some continuous conserve regions were revealed through the NP amino acid alignment of the TSWV serogroup members (Figure 2), in which the common epitope recognized by TNP MAb may be present. Therefore, we proceeded to explore this common epitope.

The full-length or truncated TSWV NPs fused with nss-tagged GFP were constructed to map the epitope recognized by TNP MAb (Figure 3A). TSWV NP (G-TNP_1-258_) was truncated into three-fourths (G-TNP_67-258_), one half (G-TNP_133-258_), and one-fourth (G-TNP_200-258_) lengths, in which the continuous conserved region was sequentially removed, and the recombinant proteins were expressed in bacteria. All GFP recombinant proteins with different length NPs (G-TNPs) were expressed in comparable levels as detected by NSscon MAb [37] or TNP MAb (this study). The results revealed that the epitope of the MAb 20C4C8 is located at the last one-fourth at the C-terminal part (aa 200-258) of NP, since all other NPs truncated before aa 200 reacted well with the Mab, but not G-TNP_200-258_ (Figure 3A,B).

According to the alignment result, the highly conserved domains of **a**, **b**, **c**, and **d** (Figure 2) did not contain the epitope reacting with the TNP MAb since they were all located upstream of aa 200. When the C-terminal part of TSWV NP_200-258_ was further truncated from its C-terminus, TNP MAb detected the recombinant proteins of G-TNP_200-244_ or G-TNP_200-229_, within which the conserved domain of “^211^KFKEYA^216^” (Figure 2) was present. Two other recombinant proteins, G-TNP_200-214_ (the conserved domain truncated) and G-TNP_218-244_ (the conserved domain deleted), were not detected by TNP MAb 20C4C8, indicating the essential role of the conserved domain as the epitope (Figure 3A,B). Since the nss-tag was tagged at the N extreme of GFP, all G-TNP recombinant proteins reacted well with NSs MAb. Their correct expression was verified as the protein sizes detected by NSscon MAb were well matched with the predicted sizes of the truncated NPs (Figure 3A,B). Although G-TNP_200-214_ and G-TNP_218-229_ did not react with TNP MAb, they did respond to NSscon MAb [37], indicating that the truncated NPs were expressed in a high quantity (Figure 3B). Thus, our epitope mapping results indicated that the epitope recognized by TNP MAb was located within aa 200–229 of TSWV NP.

### 2.3. The Amino Acid Sequence of NP_200-229_ Containing the Core Sequence KGKEYA Recognized by TNP MAb

The fragment of NP aa 200-229 containing the conserved sequence of “KGKEYA” (Figure 2) was further trimmed from its N-extreme or C-extreme to determine the minimal length of the polypeptide reacting with the MAb. The recombinant G-TNP_200-220_ responded vigorously, and G-TNP_200-215_ and G-TNP_211-220_ reacted relatively weaker with TNP MAb, but there was no reaction with G-TNP_200-214_ or G-TNP_212-229_ (Figure 4A,B). Our results indicated that the five aa residues “^210^KGKEY^215^” represent the minimal length for TNP MAb recognition. Still, its reaction was relatively weaker than that of the sequence containing six residues “^211^KGKEYA^216^” (Figure 4B). Therefore, the core sequence of “KGKEYA”, designated as the np sequence, was used for further study. Since all the expressed proteins were also carrying the nss-tag, they were detected by NSscon MAb [37] with comparable signals. The results indicated that each truncated NP peptide was expressed at similar levels (Figure 4B).

### 2.4. The Feasibility of the NP Sequence for Tagging Different Recombinant Proteins in the Bacterial Expression System

Because of the high titer and specificity of TNP MAb, the use of the np sequence as an epitope tag for monitoring recombinant proteins was attempted. In addition to GFP, the ZYMV CP [38] and a chimeric dust mite allergen Dp25 [38] were chosen to test the feasibility of tagging with the np sequence and detection by the TNP MAb in the bacterial pETsa system [38]. The expression levels of all the recombinant proteins were comparable, as shown by the western blot assay using the corresponding antibodies against individual proteins (Figure 5A). Our results revealed that the three test proteins fused with the np sequence, either at their N- or C-terminus, were readily detected by the TNP MAb. The sensitivity was comparable to the detection by the nss-tag caught with NSscon MAb [37] (Figure 5B). Remarkably, the intensities of the reaction signals with both TNP MAb and NSscon MAb were even more substantial than the corresponding polyclonal antisera in the bacteria expression system (Figure 5B).

### 2.5. Co-Immunoprecipitation of NP-Tagged or NSs-Tagged ZYMV CP In Vitro

The interaction of potyviral CP and HC-Pro is essential for the aphid transmission of potyviruses [40,41,42]. In order to test if the np sequence could be used for a co-immunoprecipitation analysis of the interacting proteins, the bacteria-expressed np-tagged ZYMV CP was mixed with the non-tagged HC-Pro protein, and the interacting complex was immunoprecipitated using TNP MAb. Our results showed that the CPs tagged with the np sequence, either at N- or C-terminus, were pulled down by TNP MAb, and the non-tagged HC-Pro was detected by the HC-Pro antiserum in the pulled down fractions, indicating its interaction with the CP (Figure 6). Previously, we have proved that the nss-tag is employed well in co-immunoprecipitation experiments to study the CP and HC-Pro interaction [38]. Our results in this investigation demonstrate that the np-tag is also applicable for co-immunoprecipitation to explore protein–protein interactions. In addition, the np-tag and the nss-tag, attached at the N- or C-extreme of the CP, do not interfere with each other.

## 3. Discussion

Orthotospoviruses are economically important, with a broad host range and wide geographic distribution. New orthotospovirus species have emerged rapidly due to global warming and thrips rampancy, causing severe crop damages worldwide. For diagnosis purposes, we have produced many polyclonal and monoclonal antibodies (MAbs) against viral proteins from different orthotospoviruses. Previously, two MAbs against the NSs proteins of TSWV and INSV, which belong to Euro-America orthotospoviruses [19], were mixed separately with NSscon MAb which recognizes the common epitope of the NSs proteins of Asia orthotospoviruses [37] and can detect all orthotospoviruses in the genus level [19]. In this study, TNP MAb 20C4C8 was generated against the NP of TSWV. It reacted strongly with five Euro-America type orthotospoviruses as revealed by western blotting but not INSV, a distinct serotype based on our previous serological classification [17] (Figure 1). The TNP MAb with a high titer of 1,024,000, which it is suggested should be used at 16,000× dilution, is a precious broad-spectrum serological tool to detect the TSWV serogroup members of orthotospoviruses.

Epitope tags are essential for detecting recombinant proteins, and they can ease the difficulty of producing specific antibodies for individual proteins. However, since protein tags can interfere with the structural and functional aspects of the fusion proteins, it is generally desirable that the protein tags are small in size to minimize any adverse effects. Previously, we used the identified core 9-aa residues of NSscon, the nss sequence of KFTMHNQIF in the NSs proteins of the WSMoV serogroup of orthotospoviruses [37] as an epitope tag for labeling various recombinant proteins expressed in different systems of bacteria, plant viral vectors, and agroinfiltration assays. The novel nss-tag is valuable for recombinant proteins, as reflected in the superior sensitivity as detected by western blotting, indirect ELISA, and co-immunoprecipitation using the NSscon MAb [38].

The five aa residues, ^211^KGKEY^215^, were identified as the minimal length recognized by TNP MAb; however, the reaction was relatively weaker compared to the np sequence containing additional A at its C extreme (Figure 4). Thus, to apply the np sequence as an epitope tag, the six residues of ^211^KGKEYA^216^ were used for tagging recombinant proteins in the bacterial expression system. When the np tag was placed at either the N- or C-terminus of GFP, ZYMV CP, and the dust mite chimeric allergen Dp25, the reaction with TNP MAb had a similar sensitivity (Figure 5).

According to the alignment result, KGKEYA is a highly conserved sequence among Euro-America type orthotospoviruses (green box in Figure 2). The sequence of INSV (KAKQYA,) was the closest to the np sequence, with only two amino acids different from the identified epitope KGKEYA, but it did not react with TNP MAb at all (Figure 1B). The sequence of WSMoV (MFKQAV) has five amino acid differences, and the sequence of IYSV (IFDETI) is completely different from the identified epitope sequence of TNP MAb. Our western blot (Figure 1B) results clearly showed that the NPs of these viruses were not reacting with TNP MAb. From the alignment and the western blotting, the specificity of TNP MAb can be concluded. Besides, when we used three very different proteins, GFP, ZYMV CP and Dp25 for tagging, the nonspecific reaction was not observed. Moreover, in Figs. 1, 2, 3, 4, and 5, the non-specific background with the host proteins was not observed. This further explains the specification of TNP MAb.

Moreover, when the nss-tag and the np-tag were fused at opposite extremes on a single protein, the detectability of both markers was not altered, and no cross-reaction between these two tags occurred (Figure 5). Thus, these two epitope tags can be efficiently used for tagging recombinant proteins for monitoring their expression, functional assay, and protein–protein interaction. Our orthotospoviral epitope tags present a novel system developed from plant pathogenic viruses. The corresponding MAbs have an outstanding contribution as a tool to promptly detect and identify the noxious orthotospoviruses.

The structure of TSWV NP is divided into three parts of N-arm, core domain, and C-arm. The N-arm and C-arm interact with the core domain of another NP monomer and form an asymmetry trimer to bind viral RNA [10]. The np sequence of _211_KGKEYA_216_ is a α helix domain and may interact with the C-arm [10], and A_216_ was predicted as an interaction residue [10]. A_216_ is also a highly conserved motif in orthotospoviral NPs, and KGKEYA is possibly an essential domain for self-interaction for polymerization and viral RNA encapsidation.

Tags are often used together to analyze the possible functions of the protein they are attached to [44,45]. Multiple affinity tags are widely used for the detection, purification [25,28], solubilization, and immobilization of recombinant proteins [26,27]. The his-tag is one of the most widely used affinity tags. The metal ion-coordinating property of the his-tag enables the purification of the his-tagged target proteins from crude extracts in a single step by metal-chelate affinity chromatography using a nickel-nitriloacetic acid (Ni-NTA) resin [29]. Hence, recombinant proteins can be engineered with orthotospoviral epitope tags and the his-tag for specific immuno-detectability by the corresponding MAb and efficient purification by Ni-NTA affinity column chromatography.

Co-immunoprecipitation is a commonly used approach to analyze protein–protein interactions. The conserved DAG motif of potyviral CP interacts with the PTK motif of potyviral HC-Pro [46,47], and this interaction is essential for aphid transmission of potyviruses [40,41,42]. As a tiny peptide of six aa, the np-tag can avoid significant conformational changes in tagged proteins. It retains a sufficient affinity to strongly bind to TNP MAb for the co-immunoprecipitation of the tagged protein with the non-tagged interacting protein(s), as reflected in the co-immunoprecipitation of np-tagged ZYMV CP. Furthermore, the complex of the bacteria-expressed np-tagged ZYMV CP and the non-tagged HC-Pro was immunocaptured by TNP MAb that was bound to protein A-coupled magnetic beads (Figure 6), indicating the potential of the np-tag to serve for protein purification in a protein A-based system.

The Co-IP results were weaker than our previous study with the NSscon Mab with the nss sequence [38]. This may be due to the fact that the nss sequence has nine amino acids, three more than the np sequence used in this study. The different effect may be due to the fact that the six amino acids of the np sequence is too short. In addition, the high specificity of the monoclonal antibody may also cause some disadvantages in such experiments. To amend this defect, the length of the np sequence with its flanking sequences in the TSWV NP should be increased, or duplicated np sequences should be used to enhance the effect of co-IP which will be further investigated.

Moreover, both in the western blotting and ELISA analyses, the MAb did not show any nonspecific cross-reactions with the host plants. The application of the np-tag and TNP MAb may have limitations because it recognizes a short six aa peptide sequence which may have some similar sequences in the other proteins also recognized by TNP MAb. However, no background was shown by western blotting while the antigen from the virus-infected plant tissue and the bacterial-expressed recombinant proteins were detected, indicating that a non-specific reaction does not happen in the tested bacteria and plant systems. Nevertheless, the feasibility of the np-tag needs to be further verified in other different expression systems such as animal and yeast systems.

In this study, we produced a valuable TNP MAb that recognized the TSWV tospovirus serogroup of Euro-America type orthotospoviruses. The common epitope of the NPs of the recognized viruses was identified through peptide mapping, and the core sequence of “KGKEYA”, designated as the np sequence, was used to tag various proteins in the bacterial expression system. Moreover, the np-tag was utilized to detect the protein–protein interaction of ZYMV CP with ZYMV HC-Pro in transient expression by agroinfiltration in host plants. Therefore, the combination of the nss-tag [38] and the np-tag in this study has established a novel tagging system for monitoring proteins and exploring protein–protein interactions.

## 4. Materials and Methods

### 4.1. Virus Sources

Eight orthotospoviruses, including the TSWV, the tomato chlorotic spot virus (TCSV), the ground ringspot virus (GRSV), the impatient necrotic spot virus (INSV), the alstroemeria necrotic streak virus (ANSV), the chrysanthemum stem necrotic virus (CSNV), the WSMoV, and the IYSV, were used for this study. TSWV was isolated from the tomato in New York (TSWV-NY) [48]. TCSV was provided by DSMZ Plant Virus Collection, Germany [49]. GRSV was collected from an infected tomato in Brazil (GRSV-BR) [50]. INSV was collected from impatiens in the United States (INSV-M) [51]. ANSV was provided by Komolink [52]. CSNV was provided by Tsuchiya [53,54]. The five viruses of TSWV, TCSV, GRSV, ANSV, and CSNV belong to the TSWV serogroup, and coupled with INSV, they are regarded as Euro-America type orthotospoviruses [19].

WSMoV was isolated from a watermelon in Taiwan [55], and IYSV was provided by R. Kormelink (Wageningen University, The Netherlands) [56]. The two viruses belong to different serogroups and are regarded as Asia type orthotospoviruses [19].

All viruses were maintained in the systemic host *N. benthamiana* Domin and the local lesion host *Chenopodium quinoa* willd by mechanical transfer under temperature-controlled (25~28 °C) greenhouse conditions. The buffer used for inocula preparation was 0.01 M potassium phosphate (pH 7.0) containing 0.01 M sodium sulfite.

### 4.2. Purification of Bacteria-Expressed TSWV NP

The open reading frame (ORF) of TSWV NP was amplified with the primer pair P-TNP-BamHI/M-TNP-XhoI (Table 1) from the construct pZTSWV-N [48] and introduced into the pET28 vector (Novagen, Sacramento, CA, USA) by *Bam*HI and *Xho*I sites to generate pET28-TNP for TSWV NP expression and purification.

The protein purification from the bacteria followed the previous approach with modifications [57]. Bacterial cells were collected 5 h after IPTG induction and used for sonication. The inclusion bodies containing the recombinant NPs were treated with a lysis buffer containing 6 M Urea and centrifuged at 15,000 rpm for 15 min. The supernatant was infiltrated through a 0.45-micron membrane and loaded into the Ni-NTA column (Novagen, Madison, WI, USA), and eluted with the buffer. The eluted proteins were separated by 12% sodium dodecyl sulfate (SDS)-polyacrylamide gel electrophoresis, visualized by soaking the gel in cold 0.05 M KCl, and eluted from the sliced gel using a Model 442 Electro-Eluter (Bio-Red, Hercules, CA, USA).

### 4.3. Production of Mouse Antisera and Monoclonal Antibodies

Six to eight-week-old female BALB/cByJ mice (Academia Sinica, Taipei, Taiwan) were individually immunized with 50 μg of the purified bacteria-expressed TSWV NP in 250 μL of phosphate-buffered saline (PBS) emulsified with an equal volume of Freund’s complete adjuvant for the first injection (Difco Laboratories, Franklin Lakes, NJ, USA), and then with an equal volume of Freund’s incomplete adjuvant (Difco Laboratories, Franklin Lakes, NJ, USA) for next two weekly injections. The mice were sacrificed 3 days after the final boost injection. The blood of the sacrificed mice was collected to prepare mouse polyclonal antiserum, and the spleen cells were harvested for cell fusion with FOX-NY myeloma cells (American Type Culture Collection, Manassas, VA, USA) following a method described previously [48]. Hybridoma cells were selected by the HAT selection medium and cultured in a 37 °C incubator supplied with 6% CO_2._ Cultured media were collected, and the secreted antibody against NP was screened by indirect enzyme-linked immunosorbent assays (indirect ELISA) [37], using crude extracts prepared from leaf tissues of TSWV-infected *N. benthamiana* plants 5 days after inoculation as the crude antigen.

The antibody-secretion hybridoma cells were cloned by limiting dilution and reconfirmed by indirect ELISA. Stable hybridoma cell lines were selected after three cycles of cloning. Pristane-primed BALB/cByJ mice were intraperitoneally injected with 1.0 × 10^6^ hybridoma cells of individually selected hybidoma lines to produce their corresponding ascitic fluids [37]. The specificity of the reaction of the selected MAbs was determined by western blotting described previously [38].

For titrating the antisera and ascitic fluids, four-fold serial dilutions starting from 2000× dilution were used for titration by an indirect ELISA [37]. The absorbance at 405 nm was determined at 30 min after the addition of the substrate.

### 4.4. Reactions of the Selected TSWV-NP MAb 20C4C8 with Different Orthotospoviruses

The serological relationships of the NPs of eight orthotospoviruses were determined by western blotting using the MAbs produced in this study. The ascitic fluids generated by a selected MAb 20C4C8 to TSWV NP, designated as TNP MAb, were used at 5000× dilution. AP-conjugated goat anti-mouse IgG (Jackson ImmunoResearch Laboratories) was used at 5000× dilution as the secondary antibody. Crude antigens extracted from leaves of healthy and individual orthotospovirus-infected *N. benthamiana* plants, including TSWV, TCSV, GRSV, ANSV, CSNV, WSMoV, INSV, and IYSV, were used for analysis.

### 4.5. Search for the Highly Conserved Domains of NPs by Sequence Alignment

Since the selected TNP MAb recognized all the TSWV serogroup members tested, the NP sequences of TCSV (Acc. No. HQ634664), GRSV (AF251271), ANSV (GQ478668), CSNV (JQ764839), INSV (D00914), WSMoV (NC003843), IYSV (NC029800), and TSWV (L12048) were collected from the NCBI databank, translated into amino acid sequences, and aligned by software BioEdit 7.2 [58]. The highly conserved domains which may have contained the common epitope recognized by TNP Mab were identified.

### 4.6. Epitope Mapping Using Full Length or Truncated TSWV NP Fused with NSs-Tagged GFP

To locate the epitope on TSWV NP recognized by TNP MAb, the complete (258 amino acids) and truncated TSWV NPs were amplified from the pZTSWV-N [48] by PCR using corresponding primer pairs as listed in Table 1. The PCR products were inserted into pETsa-nss-GFP, which expresses an nss-tagged GFP that can be monitored by NSs MAb [38], by *Bam*HI and *Kpn*I digestions, to generate the full or N-terminal-region truncated NP constructs of pETsa-nss-G-TNP_1-258_, pETsa-nss-G-TNP_67-258_, pETsa-nss-G-TNP_133-258_, and pETsa-nss-G-TNP_200-258_. The numbers indicate the actual lengths of the trimmed TSWV NPs.

The result of the N-terminal truncation described above revealed that only TNP_200-258_ reacted with TNP MAb; thus, the products were amplified by PCR from pETsa-nss-G-TNP_200-258_ using the forward primer P-T7-pro and the corresponding reverse primers (Table 1) were used to generate different C-terminal truncated NP_200-258_. The amplified NP segments were introduced into pETsa-nss-GFP by *Nco*I and *Kpn*I digestions to generate the C-terminal-region trimmed constructs of pETsa-nss-G-TNP_200-244,_ pETsa-nss-G-TNP_200-229_, and pETsa-nss-G-TNP_200-214_.

### 4.7. Identification of the Minimal Length of the Epitope

The results of the above mapping revealed that the epitope recognized by TNP MAb was located at aa 200-229 of TSWV NP. Therefore, this NP fragment was further trimmed to identify the minimal length of the epitope. The constructs of pETsa-nss-G-TNP_200-220,_ pETsa-nss-G-TNP_200-215_, and pETsa-nss-G-TNP_200-214_ deleted the aa residues from the C-terminal end, and the pETsa-nss-G-TNP_211-220_, pETsa-nss-G-TNP_212-229_, and pETsa-nss-G-TNP_218-229_ deleted the N-terminal residues. All were generated by the same approach using corresponding forward primers and the reverse primer M-pET-PsiI (Table 1) for the PCR, and the products were then cloned through the restriction sites *Bam*HI and *Psi*I.

### 4.8. Evaluation of the NP Sequence for Tagging GFP, ZYMV CP, and Mite Chimeric Allergen Dp 25 in the Bacteria-Expressed System

The nss-tagged pETsa bacterial expression vectors were modified from plasmids of pETsa-nss-GFP and pETsa-GFP-nss [38] for further fusion with the identified minimal length of the np sequence, “KGKEYA” at either the C-terminus or the N-terminus of GFP to generate double-tagged pETsa vectors, the pETsa-nss-GFP-np or pETsa-np-GFP-nss vector, respectively. The 860-bp PCR product amplified from the pETsa vector with the primer pair P-pETMluI/M-tnp-NcoI (Table 1) was introduced into pETsa-GFP-nss via the *Mlu*I and *Nco*I sites to add the np-tag at the N terminus of GFP to generate a double-tagged construct of pETsa-np-GFP-nss. The 576-bp PCR product amplified from the pETsa vector [38] by the primer pair P-tnp-KpnI/M-pET-PsiI (Table 1) was introduced into pETsa-nss-GFP via the *Kpn*I and *Xma*I sites to add the np sequence at the C terminus of GFP to generate the construct pET-nss-GFP-np. The plasmids expressing the double-tagged coat protein of the zucchini yellow mosaic virus (ZYMV CP) [38] and the dust mite chimeric allergen Dp25 [38] were constructed from the individual pETsa vectors [38] using the same primers and the same approach, to generate pETsa-np-ZCP-nss, pETsa-nss-ZCP-np, pETsa-np-Dp25-nss, and pETsa-nss-Dp25-nss.

### 4.9. Protein Expression and Detection by Western Blotting

The plasmids containing double-tagged constructs of GFP, ZYMV CP, or Dp5 were transferred into *E. coli* BL21 cells for protein expression following the manual of pET system (Novagen, Madison, WI, USA). The bacteria cells at 5.0 OD_600_ were pelleted at 14,000 rpm by microcentrifuge (Thermo Scientific™ Micro 17) for 3 min, boiled in a 200 µL sample buffer (100 mM Tris, pH 6.8, 2% SDS, 5% ß- mercaptoethanol, 15% glycerol and 0.005% bromophenol blue) for 5 min, and used for the western blot assay.

The western blot assay followed the method described previously [38]. The blottings were conducted with the primary antibody of NSscon MAb (10,000× dilution) [37], TNP MAb (10,000× dilution), GFP antiserum (4000× dilution) [39], ZYMV CP antiserum (5000× dilution) [39], or Dp5 antiserum (5000× dilution) [39]; and followed by AP-conjugated goat anti-mouse immunoglobulin (Jackson, MA, USA) or AP-conjugated goat anti-rabbit immunoglobulin (Jackson) as the secondary antibody. After the final reaction, the membranes were recorded by colorimetric detection or chemiluminescence detection.

### 4.10. Co-Immunoprecipitation of NP-Tagged Proteins In Vitro

We further examined the possibility of using the np sequence to monitor interacting proteins in a co-immunoprecipitation analysis. *E. coli* BL21 cells were transformed with the individual plasmids carrying ZYMV CP with or without double tags (pETsa-nss-ZCP-np, pETsa-np-ZCP-nss, or pETsa-ZCP). An untagged ZYMV HC-Pro sequence in the construct of pETsa-ZHC [38] was also used for this study. Following the expression of the recombinant proteins induced by IPTG, *E. Coli* cells were pelleted and resuspended in 1 mL extraction buffer (50 mM Tris-HCl, 150 mM NaCl, 0.5% TritonX-100, 5% glycerol, 1 mM EDTA, and 0.02% NaN_3_) containing a protease inhibitor cocktail (Roche Diagnostics, IN, USA), lysed with a sonicator 250-450 Sonifier Analog Cell Disruptor (Branson, CT, USA), and then centrifuged at 13,000× *g* for 5 min. The presence of recombinant proteins in soluble fractions was confirmed with the antiserum against ZYMV CP [39] or HC-Pro [43]. Aliquots of each 300 μL sample containing ZYMV CP tagged with the np sequence were mixed with each 100 μL sample containing non-tagged HC-Pro and incubated at 4 °C for one hour. Then, 25 μL Mag Protein A-Sepharose (GE Healthcare Life Sciences, Uppsala, Sweden) was added, and the mixture was incubated further for 1 h. The tubes were kept on a magnetic platform (MagRack6) to capture the Mag Protein A-Sepharose beads. After washing with a 600 µL extraction buffer two times, the beads were resuspended in a 100 µL sample 20C4C8 buffer, and the non-tagged proteins pulled down by TNP MAb were analyzed by western blotting using the antiserum against ZYMV HC-Pro [43] or CP [39] for verification.

## Figures and Tables

**Figure 1 ijms-22-08583-f001:**
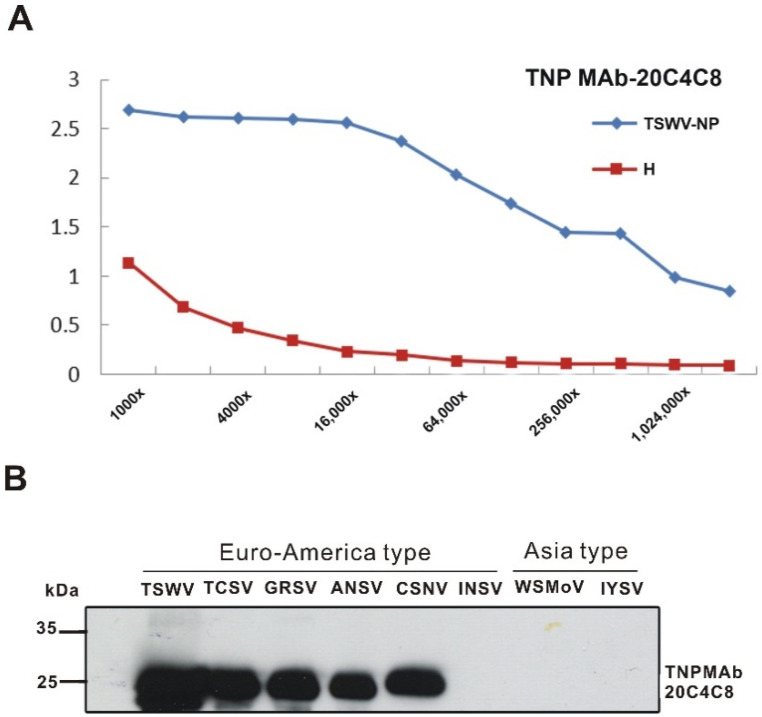
Characterization of the monoclonal antibody (MAb) 20C4C8 against the nucleocapsid protein (NP) of the tomato spotted wilt virus (TSWV), designated as TNP MAb. (**A**) Titration of the TSWV NP (TNP) MAb against the NP of TSWV. The TSWV-infected leaf tissue of a *Nicotiana benthamiana* plant 10 days after inoculation was used as the antigen source at four-fold dilution. H indicates the uninfected control. (**B**) Serological reactions of the TNP MAb with NPs of different orthotospoviruses. Detection of the NPs of orthotospoviruses by TNP MAb in western blotting. The Euro-American type orthotospoviruses used for the test included five TSWV serogroup members of TSWV, the tomato chlorotic spot virus (TCSV), the groundnut ringspot virus (GRSV), the alstroemeria necrotic streak virus (ANSV), the chrysanthemum stem necrosis virus (CSNV), and the impatiens necrotic spot virus (INSV) serotype. The Asia type orthotospoviruses tested were the watermelon silver mottle virus (WSMoV), the iris yellow spot virus (IYSV), and the type members of WSMoV and INSV serogroups, respectively.

**Figure 2 ijms-22-08583-f002:**
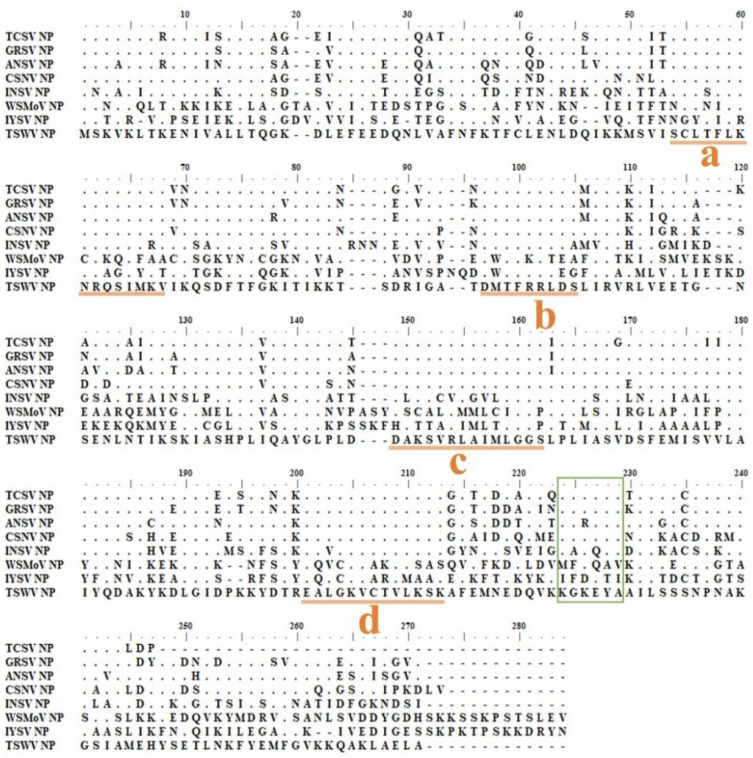
Amino acid sequence alignment of NP proteins from five TSWV serogoup members of the Euro-America type orthotospoviruses. The NP sequences of the tomato chlorotic spot virus (TCSV) (HQ634664), the groundnut ringspot virus (GRSV) (AF251271), the alstroemeria necrotic streak virus (ANSV) (GQ478668), the chrysanthemum stem necrosis virus (CSNV) (JQ764839), the impatiens necrotic spot virus (INSV) (D00914), the watermelon silver mottle virus (WSMoV) (NC003843), the iris yellow spot virus (IYSV) (NC029800) and the tomato spotted wilt virus (TSWV) (L12048) were aligned by ClustalW. Four continuous conserved domains with 100% aa identity (**a**–**d**) are indicated and underlined. A highly conserved domain in the C-terminal region is presented with a square.

**Figure 3 ijms-22-08583-f003:**
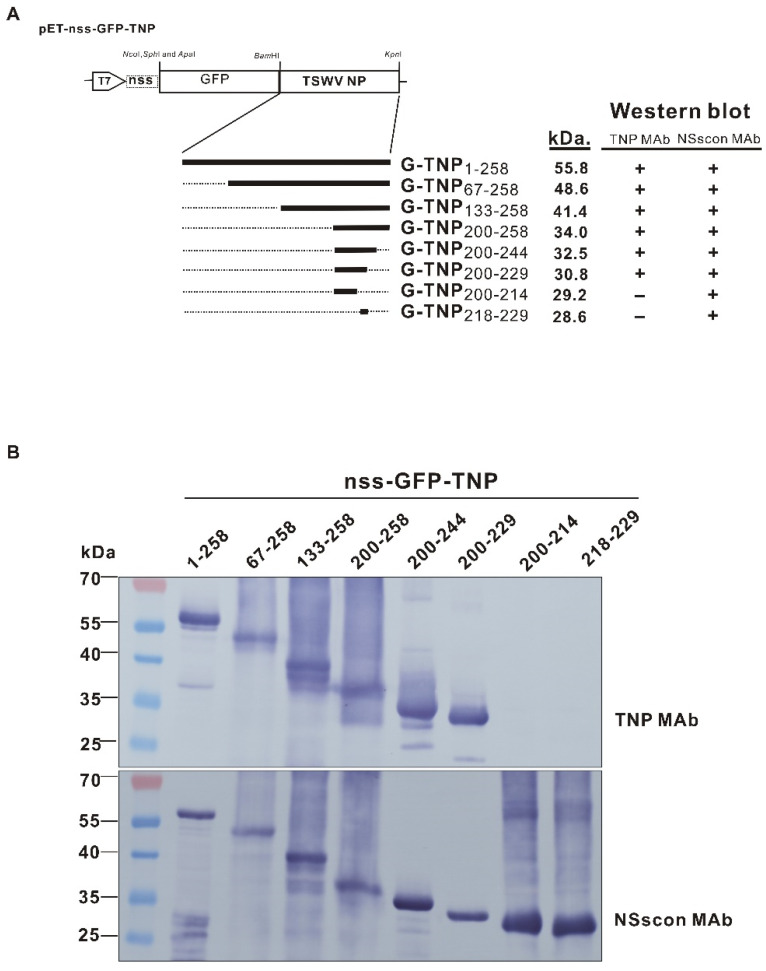
Mapping of the TSWV NP sequence recognized by the monoclonal antibody TNP MAb. (**A**) The construction of the bacteria-expressing vector, pET-nss-GFP-TNP1-258, containing the entire length (258 aa) or the various deletions of the tomato spotted wilt virus (TSWV) NP fused with the C-extreme of GFP. The thick solid lines indicate the NP sequences, and the numbers indicate the actual lengths of the deleted NPs. G-TNP_1-258_ indicates GFP fused with full-length NP. (**B**) Western blot detection of recombinant GFPs fused with various sizes of TSWV NP by TNP MAb (this study) or NSscon MAb [37]. All reactions are also indicated in panel (**A**).

**Figure 4 ijms-22-08583-f004:**
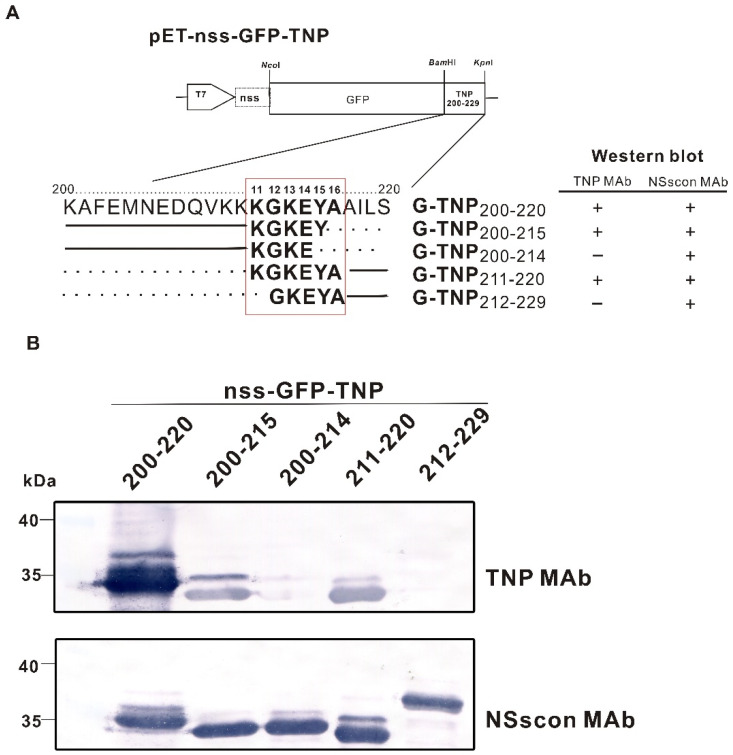
Determination of the core sequence recognized by TNP MAb against the nucleocapsid protein (NP) of the tomato spotted wilt virus (TSWV). (**A**) Constructs of the bacteria-expressing vector pET−nss−GFP−TNP, in which nss−tagged GFP was fused with various deletion sequences of TSWV NP to react with TNP MAb. The numbers indicate the amino acid positions of the NP protein. The solid lines indicate the remaining amino acids of the NP, and the dotted lines indicate the deleted amino acids. (**B**) Detection of GFPs fused with different lengths of aa 200-229 of NP by western blotting using the TNP MAb (this study) or NSscon MAb [37]. The numbers indicate the aa positions of TSWV NP. All reactions are also indicated in panel (**A**).

**Figure 5 ijms-22-08583-f005:**
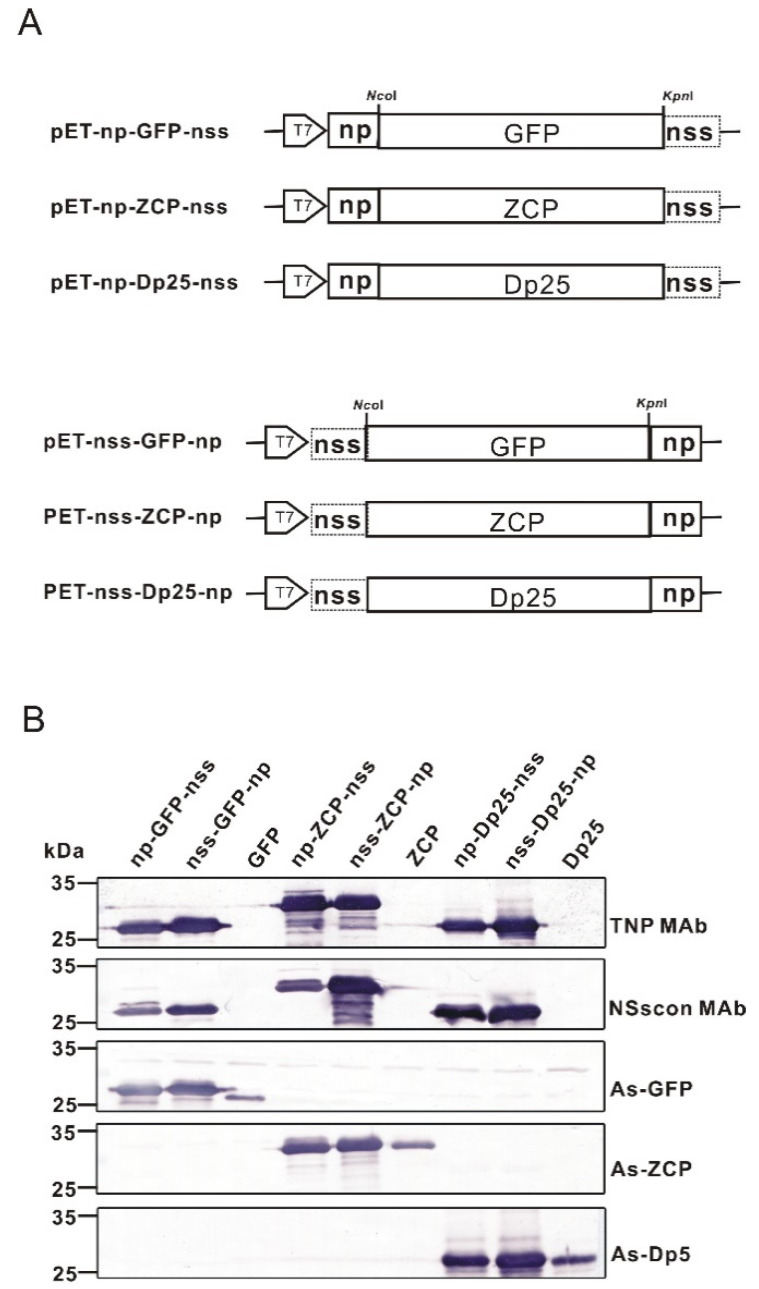
Application of the epitope core sequence KGKEYA of the tomato spot wilt virus (TSWN) NP as a protein tag in the bacteria-expression system. (**A**) Constructions of the bacteria-expressing vector in which the expressed proteins of the green fluorescence protein (GFP), the coat protein of zucchini yellow mosaic virus (ZCP), or the dust mite chimeric allergen (Dp25) were fused with the epitope core sequence KGKEYA of TSWV NP at either the N- or C-terminus. (**B**) Detection of recombinant proteins tagged with the np sequence KGKEYA by western blotting using TNP MAb (this study), NSscon MAb [37], GFP antiserum (As-GFP) [39], ZCP antiserum (As-ZCP) [39], or Dp5 antiserum (As-Dp5) [39].

**Figure 6 ijms-22-08583-f006:**
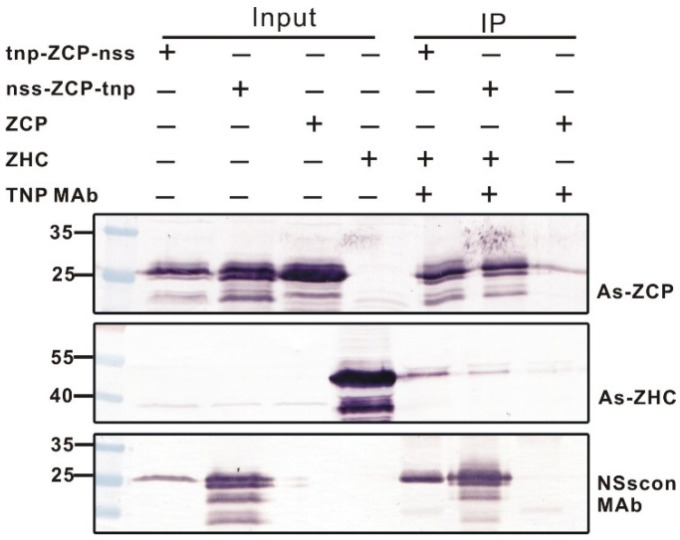
Co-immunoprecipitation of the np-tagged zucchini yellow mosaic virus (ZYMV) coat protein (ZCP) using TNP MAb 20C4C8 in vitro. The recombinant ZYMV CP and HC-Pro were expressed in np-tagged and untagged form, respectively, by the bacterial expression system. “Input” indicates that the solution contained only the np-tagged CP (ZCP) or the non-tagged HC-Pro (ZHC). “IP” indicates that the TNP MAb was used to pull down the np-tagged CP from the CP/HC-Pro mixed solutions. The presence of np-tagged CP or untagged HC-Pro from the “Input” and “IP” fractions was detected by western blotting using ZYMV CP antiserum (As-ZCP) [39], HC-Pro antiserum (As-ZHC) [43] and NSscon MAb [37], separately.

**Table 1 ijms-22-08583-t001:** Oligonucleotide primers used for this study.

Primer ^a^	Sequence (5′-3′)	Position of a.a. ^b^	Rz ^c^
**TSWV NP expression in bacteria**			
P-TNP-BamHI	AAGGGATCCATGTCTAAGGTTAAGCTCA	1	*Bam*HI
M-TNP-XhoI	CACTCGAGAGCAAGTTCTGTGAGTT	258	*XhoI*
**Epitope mapping of TSWV NP**			
P-T7-pro.	TAATACGACTCACTATAGG	V	*Nco*I
P-TNP-199	AAGGGATCCATTAAGCAAAGTGATTTTACT	67	*Bam*HI
P-TNP-397	AAGGGATCCCCTCTTGATGATGCAAAGT	133	*Bam*HI
P-TNP-598	AAGGGATCCAAAGCATTTGAAATGAATG	200	*Bam*HI
P-TNP-634	AGGGATCCGGAAAAGAGTATGCTGCTAT	212	*Bam*HI
P-TNP-637	AGGGATCCAAAGAGTATGCTGCTATACT	211	*Bam*HI
P-TNP-652	GGGATCCATACTTAGCTCCAGCAATCC	218	*Bam*HI
M-TNP-KpnI	CAGGTACCAGCAAGTTCTGTGAGTT	258	*Kpn*I
M-TNP-732	CAGGTACCTTCATAGAACTTGTTAAGAGTTTC	244	*Kpn*I
M-TNP-687	CAGGTACCACTTCCTTTAGCATTAGGATTG	229	*Kpn*I
M-TNP-642	CAGGTACCCTCTTTTCCTTTCTTCACCTG	214	*Kpn*I
M-TNP-654	CAGGTACCTATAGCAGCATACTCTTTTCCT	218	*Kpn*I
M-TNP-660	CAGGTACCGCTAAGTATAGCAGCATACTC	220	*Kpn*I
M-TNP-645	CAGGTACCATACTCTTTTCCTTTCTTCACC	215	*Kpn*I
M-pET-PsiI	AAAATCCCTTATAAATCAAAAGAAT	V	*Psi*I
**np sequence tagging** ^**d**^			
P-tnp-KpnI	GATCCGGTACC*AAAGGAAAAGAGTATGCT*TGACTCGAGCACCACCACCA	211-216	*Kpn*I
M-pET-PsiI	AAAATCCCTTATAAATCAAAAGAAT	V	*Psi*I
P-pET-MluI	CAGCCCACTGACGCGTTGCGCGA	V	*Mlu*I
M-tnp-NcoI	ACGCATGCCATGG*CATACTCTTTTCCTTT*CATGGTATATCTCCTTCTTA	211-216	*Nco*I

^a^ The P indicates forward primer and M indicates reverse primers. ^b^ The numbers indicate the positions of the amino acid (aa) in the nucleocapsid protein (NP) of tomato spotted wilt virus (TSWV). The V indicates primers complementary to vector sequences. ^c^ Rz indicates the restriction enzyme sites for cloning. ^d^ The np-tag sequence is indicated in italic.

## Data Availability

Not applicable.

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
