# Peer review of "Identification of a Common Epitope in Nucleocapsid Proteins of Euro-America Orthotospoviruses and Its Application for Tagging Proteins"

_ijms, 2021, doi:10.3390/ijms22168583_

Round 1

Reviewer 1 Report

The manuscript is suitable for publication in Molecular Sciences after minor changes. It should be considered as short communication. Figure 1, 2 and 3 can be grouped as a single figure.

Author Response

Dear Editor

We highly appreciate the time and effort to provide your valuable comments on the manuscript to make the MS “Identification of a common epitope in nucleocapsid proteins of Euro-America orthotospoviruses and its application for tagging proteins” more complete. We feel that Fig. 1, 2 and 3 have different purposes and contributions and would like to keep as it is.

Point1: The manuscript is suitable for publication in Molecular Sciences after minor changes. It should be considered as short communication. Figure 1, 2 and 3 can be grouped as a single figure.

Response1: We appreciate the affirmation and suggestion for the manuscript. High titer of TNP MAb and its broad-spectrum detection of TNP for TSWV serogroup Euro-America type orthotospoviruses, but no reaction with Asia-type virus, is shown in Figure 1.  The results indicate that TNP Mab is a previous tool for prompt and accurate diagnosis and identification of the worldwide noxious orthotospoviruses, thus contributing significantly.  Figure 2 indicates the possible TNP MAb-recognizing regions by sequence alignment and the actual region for the following peptide mapping experiments. This Figure 2 also has been revised according to Reviewer 2 to emphasize the importance of the possible location of the epitope. The peptide mapping of Figure 3 was designed based on the results of Fig. 2, and using various truncation of TSWV NP to identify the epitope in the C-terminal conserved region.  These three figures have the necessity of independent meaning.  We feel that they should be presented as it is.

Reviewer 2 Report

Major revisions

  1. Amino acid sequence alignment of INSV, WSMoV and IYSV should be included in fig 2.
  2. You should provide the additional experiment of point mutation(s) (substitution) on “KGKEYA” in pET-nss-GFP-TNP1-258 contain to demonstrate the specificity of TNP MAb on “KGKEYA” in fig 3.
  3. Synthetic peptide of “KGKEYA” should be involved to do competition for the specificity confirmation in fig 6.
  4. On fig 6, co-IP effect is much weak (weak signals on IP lane 1, 2 and faint signal on IP lane 3) than previous study (ref.38). It is hard to claim its functionality on co-IP. You should provide the additional experiment of nss-GFP-tnp as negative control to verify specificity.

Minor revisions

1. You should delete “Fig. 3 &” from the sentence “the reaction is relatively weaker than the np sequence containing additional A at its C extreme (Fig. 3 & 4).” on line 272-273.

Author Response

Dear Editor

We appreciate very much the time and effort that the editor and reviewers have dedicated to provide valuable comments to make the manuscript more complete. We have incorporated all suggestions provided by the reviewers into the revised version.  We have marked the changes within the manuscript and a point-by-point response list for your convenience to check the changes.

Point 1: Amino acid sequence alignment of INSV, WSMoV and IYSV should be included in fig 2.

Response1: We thank the suggestion of the reviewer. INSV WSMoV and IYSV were all included in the new sequence alignment as shown in Figure 2, and the new alignment has been placed in the revised version (line 163).

Point2: You should provide the additional experiment of point mutation(s) (substitution) on “KGKEYA” in pET-nss-GFP-TNP1-258 contain to demonstrate the specificity of TNP MAb on “KGKEYA” in fig 3.

Response2: According to the alignment result, KGKEYA is a highly conserved sequence among Euro-America type orthotospoviruses (green box in Figure2).  The sequence of INSV (KAKQYA,) is the closest to the np sequence, with only two amino acids different from the identified epitope KGKEYA, but does not react with TNP Mab at all (Fig. 1B).  The sequence of WSMoV (MFKQAV) has five amino acid differences, and the sequence of IYSV (IFDETI) is completely different from the identified epitope sequence of TNP MAb.  Our western blot (Fig.1B) results clearly showed that the NPs of these viruses were not reacting to TNP Mab.  From the alignment and the western blotting, it can conclude the specificity of TNP Mab.  Besides, we used three very different proteins, GFP, ZYMV CP and Dp25 for tagging, the nonspecific reaction was not observed.  Moreover, in Fig. 1, 2, 3, 4, & 5, non-specific background with host proteins was not observed.  This further explains the specification of TNP Mab.  Thus, we do not feel it is necessary to do substitution or mutation to confirm the specificity of “KGKEYA” with TNP Mab.  Nevertheless, we have added the above explanation in the Discussion (line 281-291 and marked green) to ease the reviewer’s concern.

Point3: Synthetic peptide of “KGKEYA” should be involved to do competition for the specificity confirmation in fig 6.

Response3: The use of synthetic peptides of KGKEYA for competition experiments in Co-IP is a good strategy.  However, KGKEYA was already confirmed as the epitope of the monoclonal antibody TNP Mab. Its specificity is shown in Fig. 1~5, so it is unnecessary to ensure the specificity issue through such a competition experiment.

Point4: On fig 6, co-IP effect is much weak (weak signals on IP lane 1, 2 and faint signal on IP lane 3) than previous study (ref.38). It is hard to claim its functionality on co-IP. You should provide the additional experiment of nss-GFP-tnp as negative control to verify specificity.

Response4: The Co-IP results was weaker than our previous study with the NSscon Mab with the nss sequence (Ref.38).  This may be due to the reason that the nss sequence has nine amino acids, three more than the np sequence used in this study.  The different effect may be resulted from that the 6 amino acids of the np sequence is too short. In addition, the highly specificity of monoclonal antibody may also cause some disadvantages in such experiments. To amend this defect, increase the length of the np sequence with the its flanking sequences in the TSWV NP or use duplicated np sequences to enhance the effect of co-IP will be further investigated. We thank the comment of the reviewer and has added the above explanation in Discussion (line 322-328 and marked green).

Minor revisions

Point1: You should delete “Fig. 3 &” from the sentence “the reaction is relatively weaker than the np sequence containing additional A at its C extreme (Fig. 3 & 4).” on line 272-273.

Response1: Reply: It was revised as suggested and the sentence was “the reaction is relatively weaker than the np sequence containing additional A at its C extreme (Fig. 4).” in line 275-276 and marked green. 

Round 2

Reviewer 2 Report

  1. Point mutation(s) (substitution) on “KGKEYA” in pET-nss-GFP-TNP1-258 will provide a direct evidence to demonstrate TNP MAb targeting on “KGKEYA”. Others you mentioned are indirect evidences.
  2. Synthetic peptide of “KGKEYA” for competition and the additional experiment of nss-GFP-tnp as a critical negative control should be involved in fig 6 for the specificity confirmation of co-IP.